# Acute and Chronic Systemic Inflammation: Features and Differences in the Pathogenesis, and Integral Criteria for Verification and Differentiation

**DOI:** 10.3390/ijms24021144

**Published:** 2023-01-06

**Authors:** Natalya Zotova, Yulia Zhuravleva, Valeriy Chereshnev, Evgenii Gusev

**Affiliations:** Institute of Immunology and Physiology of Ural Branch of the Russian Academy of Sciences, 620049 Ekaterinburg, Russia

**Keywords:** systemic inflammation, inflammatory biomarkers, sepsis, trauma, autoimmune diseases, end-stage renal disease, chronic limb threatening ischemia

## Abstract

Currently, there is rationale for separating the systemic manifestations of classical inflammation from systemic inflammation (SI) itself as an independent form of the general pathological process underlying the pathogenesis of the most severe acute and chronic diseases. With this aim in view, we used integral scales of acute and chronic SI (ChSI), including the following blood plasma parameters: interleukins 6, 8, 10; tumor necrosis factor alpha; C-reactive protein; D-dimer; cortisol; troponin I; myoglobin. The presence of multiple organ dysfunction according to the SOFA score was also taken into account. The effectiveness of the scales was tested in groups of intensive care patients during different periods of acute trauma, sepsis, and septic shock. The ChSI scale was applicable under systemic autoimmune diseases, chronic purulent infections, chronic limb threatening ischemia, and end-stage renal disease of various genesis. The number of examined patients was 764 in total. The scales allowed us to verify specific phases of acute SI and identify pathogenetic risk factors of lethal outcomes, as well as the most severe variants of the chronic pathologies course. These scales are open adaptable systems (in terms of the nomenclature and choice of indicators). They are primarily intended for scientific research. However, the SI verification methodology presented in this paper may be useful for developing advanced criteria for assessing both the typical links in the pathogenesis of many diseases and the severity of the overall condition of patients for clinical practice.

## 1. Introduction

It is now obvious that inflammatory mechanisms underlie the pathogenesis of most acute and chronic diseases [1]. At the same time, modern interpretations of the concept of “inflammation” go far beyond the canonical ideas about this general pathological process [2]. Thus, in recent years we have seen the advent of the terms “systemic inflammation” —SI (about 78 thousand mentions in the PubMed database), “systemic inflammatory response”—SIR (about 37 thousand mentions in the PubMed database), and “chronic low-grade inflammation”—ChLGI (more than 4.5 thousand mentions in the PubMed database). However, there are just a few publications in the scientific literature which analyze different variants of non-classical inflammation from the perspective of general pathology, that is, as independent types of general pathological processes. At the same time, SIR and SI are usually used synonymously, which is a fundamental error in our opinion.

A feature of classical inflammation is a focal character of microvascular exudative reaction with directional migration of leukocytes to the zone of local damage. At the same time, SIR prioritizes the task of resource supply of the inflammation focus and is usually of an adaptive nature. Systemic low-grade inflammation is characteristic of aging, morbid obesity, metabolic syndrome and type 2 diabetes [1] The development of SIR in this case is moderate, largely due to metabolic factors of damage, but without any connection with the focus of inflammation [3]. In turn, systemic hyperinflammation or SI itself are maladaptation responses to severe systemic damage of different nature, leading to the phenomenon of systemic “inflammatory microcirculation”. Such responses are characterized by a high degree lability of many homeostasis parameters at the systemic level, up to the development of the phenomenon of “cytokine storm”. The elementary functional unit of all inflammation variants, as well as many other pathological and even extreme physiological processes, is pro-inflammatory cellular stress [2], which has a number of typical (inherent in cells of different types) common links. A characteristic feature of cellular stress is the formation of an inflammatory receptor and secretory phenotype, which ultimately determines changes at the organ and organism level (Figure 1).

Thus, SIR (including the accumulation of cytokines and other inflammatory mediators in the blood, leukogram changes, activation of the hypothalamic-pituitary-adrenal system (HPA), acute phase liver response, etc.) is a very broad phenomenon covering not only pathological processes of different pathogenetic and clinical significance, but even some extreme physiological processes [1,4]. Many of the acute critical conditions and the most severe variants of chronic diseases are based on systemic microcirculatory disorders, which are closely associated with pro-inflammatory reactions of microvessels and perivascular connective tissue, activation of intravascular leukocytes, platelets, and inflammation-related plasma protein systems [5,6,7,8]. Consequences of these changes include secondary damage and dysfunction of vital organs, which is the main cause of fatal outcomes in medical institutions of different specialisms [9]. These conditions can be exactly characterized from the position of SI as an independent form of the general pathological process [2,10]. The principal difference between SI and two other major general pathological processes—tumor growth and classical inflammation—is that the presence of SI does not tell us whether the nosologies belong to the category of tumor or inflammatory diseases, being a severe complication of diseases of various genesis. Thus, SI usually develops on the back of other general pathological processes, primarily classical inflammation, which has a local focus and, as a rule, non-critical manifestations of SIR. Another distinctive feature of SI is that it is a complex phase-specific process including SIR of certain intensity, critical multiple organ dysfunction (MOD), systemic tissue alteration, and clinical signs of microcirculatory disorders, including refractory shock and disseminated intravascular coagulation (DIC). This necessitates the use of integral criteria to verify specific SIR levels and SI phase dynamics.

The third international consensus definition of sepsis and septic shock (“Sepsis-3”) redefined sepsis as “life-threatening organ dysfunction caused by an unregulated host response to infection” [11]. The updated definitions of “Sepsis-3” excluded systemic inflammatory response syndrome (SIRS) criteria, due to their low specificity to the development of critical conditions, and identified organ dysfunction as the main criterion, which can be quantified using the Sequential Organ Failure Assessment (SOFA) scale and its screening version-quick SOFA score (qSOFA). However, the problem, in our opinion, was not that the SIRS concept (“Sepsis 1” and “2”) overestimated the importance of “inflammation” in the pathogenesis of sepsis, but that systemic inflammation as the pathogenetic basis of critical conditions of infectious and noninfectious nature has not yet been characterized from the position of general pathology [12]. 

The solution to this problem cannot be just technical; rather, it requires new theoretical and methodological approaches. From the theoretical point of view, it is reasonable to divide “inflammation” into at least three “large” and independent general pathological processes: classical (canonical) inflammation, ChLGI, and SI [1,2]. From the methodological point of view, there is a need to specify and more clearly characterize SIR, subdivide SIR into several conditional levels of pathogenetic and clinical significance, and to determine the relationship between SIR and other SI phenomena [10,13].

The aim of the present work is to show the possibility of verification and differentiation of acute and chronic SI, as well as individual phases of this general pathological process using integral criteria that are available in clinical and experimental studies.

To achieve this aim, we used two variants of integral SI criteria, such as a SI scale for the verification of acute SI phases and a ChSI scale for the verification of the chronic SI variant. Both of these scales include a more specific scale of SIR levels (systemic reactivity levels—RL)—the RL scale, while the SI scale includes also the SOFA score, which is universal for the assessment of MOD in clinical practice.

The study sought to address the following objectives:To check the quality of the SI scale by comparing the results of its use with the clinical equivalents of SI, which can be ranked in the following order by the probability of SI development: refractory shock > lethal outcomes in intensive care units (ICU) > MOD without lethal outcomes in ICU patients > MOD in patients of other departments;To confirm the general pathological, multi-etiological nature of the acute SI variant by comparing acute critical conditions of infectious (sepsis) and non-infectious genesis (acute polytrauma);To verify separate phases of acute SI by comparing the hyperergic (with extremely high levels of SIR) and hypoergic (with relatively moderate manifestations of SIR) variants of SI, taking into account the time of process development and the degree of its criticality;To compare the prognostic value of the SI and RL scales and the clinical scores of polytrauma, such as APACHE-II (Acute Physiology and Chronic Health Evaluation II), TRISS (Trauma Score and Injury Severity Score), ISS (Injury Severity Score), GCS (Glasgow Coma Scale), SOFA (Sequential Organ Failure Assessment), TISS (Therapeutic Intervention Scoring System), SAPS II (Simplified Acute Physiology Score II), and criteria of the SIR syndrome (SIRS) for lethal outcomes;To evaluate the probability of SI presence in chronic pathologies of different genesis (infectious, autoimmune, endocrine, ischemic gangrene and others) depending on the severity of the disease and peculiarities of its pathogenesis;To assess the possibility of differentiation between acute and chronic SI by means of detecting false-positive results of acute SI scale application in patients with chronic diseases;To elucidate differences and similarities between certain phenomena of acute and chronic systemic inflammation.

The main significance of the presented article is to substantiate the necessity of new approaches to the understanding of the pathogenesis of critical conditions from the standpoint of general pathology. At the same time, models of general pathological processes should become the methodological basis for solving private clinical problems.

### 1.1. Methodological Approaches Used in the Work

The criteria of the general pathological process are not a tool for the validation of specific clinical definitions (nosologies and syndromes). The general pathological process model is aimed at identifying universal mechanisms of pathogenesis underlying various diseases. These mechanisms can be taken into account in the design of various clinical protocols for many diseases and syndromes.

The specification and characterization of SIR is of key importance for the verification of SI. Under the “broad approach”, almost any systemic changes in the body related in one way or another to inflammation can be attributed to SIR. However, this approach does not allow SIR to be specified. In the “narrow sense”, SIR can be defined as the accumulation of acute-phase proteins, cytokines, soluble forms of cellular stress activation receptors, and some other inflammatory mediators in blood [10,13]. Further on this basis, the most informative molecular factors of SIR can be identified and specific ranges of changes in their concentrations in blood that have an independent pathogenetic and diagnostic quality can be determined. In general, at least 100 individual SIR factors suitable for SI characterization can currently be distinguished [13]. However, changes in the concentrations of SIR factors in blood are usually characterized by non-linearity and non-normal distribution in patient groups studied, along with a low level of correlation between each other. This circumstance makes it doubtful that a “golden” (universal) marker of SIR could be found even within one nosology [13] and predetermines the need for a more detailed characterization of SIR (especially in scientific studies) using integral indices [10,13]. It should be noted that quantitative changes in individual SIR indices that are comparable in extent may have unequal pathogenetic and diagnostic significance and characterize different reactivity levels (RL) of the organism.

In our work, we identify six (from 0 to 5) principal RLs characterizing the SIR, namely: 

RL-0 characterizes the reference values of the norm;

RL-1 confirms ChLGI and SIR manifestations in classical acute and chronic inflammation, but excludes SI development;

RL-2 is typical for severe acute purulent inflammatory processes of classical type, but is also possible in some variants of depressed SI phase;

RL-3 is an overlapping zone of SI of classical acute purulent inflammatory process and systemic inflammation;

RL-4 is characteristic of hyperergic variant of SI, the probability of classical inflammation is low;

RL-5 confirms the presence of SI.

Thus, RL-0 (normal) and RL-1 should practically exclude the presence of acute SI, RL-5 confirms the presence of hyperergic SI, and RL-2-4 are probabilistic and require additional criteria to confirm SI. RL-3-5 in chronic processes, in our opinion, a priori confirm chronic systemic inflammation (ChSI), while the verification of ChSI at lower SIR values (RL-1-2) requires additional criteria.

To determine specific RL values, we used an RL scale based on the determination of five SIR factors in patient plasma, which are the following: four cytokines (TNFα, IL-6, IL-8, IL-10) and one acute-phase C-reactive protein (CRP) [10,14]. The ranges corresponding to specific RLs were individually set for each mediator (Table 1). Then, in each patient, two indicators with the lowest RLs were excluded (if there were identical RLs, three values out of five were also kept on), and the values of the three remaining SIR factors were averaged to whole RL values (0–5). This approach made it possible to not only obtain relatively stable principal SIR values (usually with a normal distribution) but also adapt this system to the specific features of SIR in each individual patient. The RL scale, in addition, is an open system with the possibility of changes in the quantitative and qualitative composition of the molecular factors used in it.

Additional criteria are needed not only for a more reliable verification of SI, but also for a comprehensive pathogenetic characterization of this complex process. To assess acute SI for this purpose, we determined, in addition to RL values, the presence of four additional phenomena (Table 2), including hypothalamic-pituitary-adrenal distress response (HPA), microthrombosis, systemic alteration, and MOD.

A separate and very important aspect of SI characterization is the determination of the dynamics of this process, i.e., its developmental phases. To detect the SI phases and the boundary state (pre-SI), we used the ratio of individual parameters in the SI scale, the time of recording/onset of the critical state, and the presence of shock (Table 3).

To verify chronic SI (ChSI), we used the same indices (excluding the SOFA score), though in a “softer” version of their values in the form of a ChSI scale (Table 4). This approach allowed us not only to compare acute and chronic SI variants in more detail but also to define the possibility of false-positive results using the SI scale in patients with chronic pathologies.

Thus, the integral criteria used enable us not only to verify the presence of SI but also to describe its pathogenetic image, including the structure of separate phenomena (their specific combination) and determination of phases of SI development. The methodology of SI verification is based on the principles of systems approach and analysis, and the general theory of fuzzy-logic and nonlinear systems [15].

### 1.2. Validation Criteria for SI Scales

#### 1.2.1. Agreement of the Results Obtained Using the SI Scale and the ChSI Scale with the Clinical Equivalents of SI

An analysis of apparent SI manifestations was performed on two cohorts of patients with refractory septic shock (SS) that developed in acute sepsis and in tertiary peritonitis (prolonged and subacute sepsis). These cohorts were characterized by different SIR severity, which makes it possible to assess the probability of verifying various SI phases using the SI scale;Sepsis and acute polytrauma (without a confirmed infectious complication) were considered as pathologies with a high risk of SI development; patients were examined on the 1st–2nd and 5th–7th days from the onset of the critical process. A comparison of the groups with infections and aseptic critical conditions points to the general pathological nature of SI;Systemic microcirculatory disorders (according to autopsies) are the main cause of mortality in intensive care units. Therefore, a separate aspect of the work was to analyze the relationship between the SI scale results and lethal outcomes. In this case, complications associated with thromboembolisms (according to autopsies), including pulmonary embolisms, secondary myocardial infarctions and strokes, were excluded from the study;The comparison group included patients with acute phlegmonas of the lower limbs meeting Sepsis-3 criteria (the presence of the infection focus and MOD) but were treated in a surgical department (without intensive care) and had no lethal outcomes. By the 5th–7th day of surgical and etiological treatment, signs of MOD had resolved in all patients in this cohort. The expected outcome of SI scale testing in this group was a relatively low detection rate of SI, especially of the most life-critical phases of SI;The efficacy of the ChSI scale was assessed by examining independent cohorts with 16 chronic pathologies of different genesis, such as infections, autoimmune and endocrine diseases, chronic organ failure, and chronic ischemia of the lower limbs. Patients in these groups had different levels of SIR and condition severity. The focus was on the analysis of the registration rates of certain phenomena of ChSI depending on the severity of the pathological process and the nature of the damaging factors. A particular aim of this study was to evaluate the specificity of the SI scale (designed for the acute variant of SI) by identifying false-positive results of this scale in chronic pathologies.

#### 1.2.2. Clinical Material Collection

The clinical material was collected at nine medical centers, and the verification of clinical diagnoses was carried out by the medical specialists of these clinics, independently of the research group.

#### 1.2.3. Laboratory Analysis of Biomarkers

Various laboratory methods can be used to measure the biomarkers included in the calculation of the SI and ChSI scales. However, the same analytical method was applied in the work in all cases, namely immunochemiluminescent analysis using the Immulite system. All of the parameters included in the SI and ChSI scales were the same (except for the SOFA scale measured only in acute SI), which provided a more correct comparison of acute and chronic SI variants.

#### 1.2.4. Selection of Biomarkers

All parameters included in the SI and ChSI scales are widely used in clinical practice. To adapt these markers to the scales, we used our methods registered in several patents for invention of the Russian Federation (No. 2004126977, No. 2005108805, No. 2005108376, No. 2005109121, No. 2006124894). In separate studies, we tested procalcitonin as an additional criterion of the SI scale. We showed that procalcitonin can be employed to verify SI not only of septic but also aseptic nature [16]. The present study does not include procalcitonin because it was measured only in some patients in the above-mentioned groups, primarily to monitor the efficacy of antibiotic therapy. Moreover, according to our data, other SI parameters, such as sIL-2R (sCD25), sE-SL (sCD62E), sICAM-1 (sCD54), and IL-1β, transforming growth factor beta 1 (TGFβ1), lipopolysaccharide-binding protein (LBP), eosinophil cationic protein (ECP), soluble fibrin, aminotransferases, etc., may be included as additional criteria in the SI scale. Of course, this is not a complete list of potential SI markers. However, in accordance with Section 1.2.3, our data about additional SI criteria are not presented in this study.

#### 1.2.5. Mathematical Analysis

We used various methods of mathematical analysis: factor, discriminant, dispersion analysis, fuzzy logic (fuzzy sets) methods, and various methods of image recognition, including the method of multilayer neural networks, to create SI and ChSI scales [17]. However, the scales we use, similarly to most other criteria in medicine, are an expert system, since we also used heuristic methods based on an analysis of the scientific literature data to create them.

#### 1.2.6. Therapy

Patients in the studied cohorts were treated according to the clinical protocols. Patients receiving specific anti-inflammatory therapy (including anti-cytokine therapy) were excluded from the study. That said, we took into account the fact that the analysis of the effect of specific therapies on SI phenomena is not a necessary task for the characterization of the general pathological process.

## 2. Results

### 2.1. Phenomena of the Acute SI Variant 

Table 5 presents the results of RL calculation and registration of SI phenomena in the studied groups while Table 6 includes specific SI phases identified; and fatal outcome, SI detection, and pre-SI rates (3–4 points of SI scale). The empirical values of the studied indices are given in the Appendix A. 

The data presented in Table 5 and Table 6 can be briefly summarized as follows:A comparison of two cohorts of patients with refractory septic shock (groups of patients with acute sepsis and tertiary peritonitis) showed that SS can proceed in two alternative scenarios (Table 5 and Table 6). The first, i.e., the hyperergic variant, predominates in the groups with acute sepsis and is characterized by RL-4-5. The second, i.e., the hypoergic variant (RL-2-3), predominates in patients with tertiary peritonitis. Lethality in these cohorts was 71% and 94%, respectively (Table 6). These two sepsis variants reflect different SI phases: the phlogogenic stroke phase (cytokine storm) and the depressive (depletion) phase. Meanwhile, the SI scale, despite these differences, confirms SI in both variants of SS in 100% of cases. Thus, the pathogenesis of various SS variants can be a priori considered in terms of SI.In contrast to patients with SS, lethality in the groups of intensive care patients with MOD (trauma and sepsis) on days 1–2 and 5–7 was significantly less (21–50%) and SI was not detected in 100% of patients (except for the group “Sepsis 5–7”, in which SI detection reached 100%). In isolated cases (4.4% in acute sepsis and 1.9% in acute trauma), neither SI nor pre-SI was detected on day 1–2 in these patients. A comparative analysis of the groups with sepsis and polytrauma showed a fundamental similarity in the patterns of SI development, but also some differences (Table 5). In particular, the greatest differences between the polytrauma and sepsis groups were found on days 5–7 (Table 5 and Table 6). Thus, in acute trauma, only 55.6% of patients (10 out of 18, Table 6) had SI on days 5–7. However, all nine fatalities in “Multiple injuries 5–7” were associated with SI. The other eight patients (without SI) had a relatively mild manifestation of MOD (SOFA score ≤ 4). All of these patients had not developed SI criteria by day 5–7 and subsequently recovered. Thus, despite some differences, the fundamental patterns of SI development in sepsis and trauma were quite comparable: in both nosologies, the presence of SI and its individual phenomena determined the severity of patients’ condition and mortality in intensive care units.Patients with “non-resuscitative” sepsis (diagnosing sepsis was made according to Sepsis-3 criteria) (comparison group) had relatively moderate manifestations of MOD (mean SOFA score—3.6) on days 1–2 from hospitalization, which resolved in all patients by the 5th–7th days (Table 5 and Table 6). This group was characterized by the development of a relatively noncritical SI phase (development phase) in 7.5% of cases; in 18.2% of patients, SIR manifested itself only as classical inflammation; and 65% of patients had a borderline pre-SI state. The intensity of SIR in the patients of this group did not exceed RL-3. However, in two patients (5%) without signs of myocardial infarction, troponin I was detected in plasma: 16.1 ng/mL and 16.2 ng/mL. In these cases, we could assume the presence of organ damage associated with relatively hidden microcirculatory disorders. In general, these data once again confirmed the correlation of SI with the criticality of the patients’ condition.

### 2.2. Association of Lethality with Specific Phases of SI

Not all phases of SI are fatal in the context of resuscitation interventions. This suggests the possibility of diagnosing relatively compensated (sub-compensated) phases of SI for timely intensive care. In addition, the detection of hyperergic phases (primary and secondary phlogogenic strokes) is indicative of the need for more active use of methods of anti-inflammatory and immunosuppressive therapy.

The rates of SI phases observed in patients with acute conditions are presented in Figure 2, Figure 3 and Figure 4:(a)The detection of SI phases (%) in all of the groups of acute pathologies mentioned above (Figure 2);(b)The rate of lethality in each SI phase (Figure 3);(c)The specific contribution of each SI phase to the total mortality of the groups (Figure 4).

Three critical phases of SI (primary and secondary phlogogenic stroke (cytokine storm), and depressive phase) were detected in 37% of cases, but the total contribution of these phases to the total mortality was 75%. The development/transition phase was found in 31% of “acute” patients and in 21% of fatalities, and these frequencies correspond to 26% and 4% for pre-SI. All cases with SIR of classical inflammation (6%) ended in recovery. At the same time, it should be noted that the development phase and the pre-SI state were a risk zone for switching to other, more severe phases of SI. In general, lethality in the verification of critical phases of SI (Figure 3) was 45%, 68%, and 95% in the series: primary phlogogenic stroke phase < secondary phlogogenic stroke < depressive phase (depletion phase). The differences in lethality between all SI phases were statistically significant (Chi-square test, *p* < 0.05).

### 2.3. Comparative Prognostic Value of SI Scale and Some Clinical Scales for Fatal Outcomes

The prognostic values (for fatal outcomes) of the SI scale and the RL scale were compared with clinical scales assessing organ dysfunction, condition severity and severity of injury in polytrauma by ROC-analyses; the results are presented in Table 7. The clinical scales studied included universal scales assessing the severity of condition-APACHE II, TISS, SAPS II, organ failure-SOFA, impaired consciousness-GCS; as well as injury-specific scales: the severity of injury-TRISS and ISS. The ISS scale is morphological, while TRISS is a combined scale (it includes patient age, ISS scale, and functional RTS scale-Revised Trauma Score). The prognostic value of the SIRS criteria (0–4) was also evaluated [18]. The APACHE II, TRISS, ISS, TISS, SAPS II severity scales were calculated only in a subset of patients who had acute lung injury (ALI) or acute respiratory distress syndrome (ARDS), so the sample of patients in this study was limited (n = 37). We included in this cohort not only 23 intensive care patients with MOD, but also fourteen intensive care patients with respiratory insufficiency alone on day 1 of the study, of whom nine were lethal.

The results of ROC-analysis showed (Table 7) that the SI scales were as good as, if not better than, the clinical scales SOFA, GCS, TRISS, and ISS. No significant results were obtained for the APACHE II, TISS, SAPS II, and SIRS scales (*p* > 0.05). ROC-analysis data are presented using MedCalc software. However, it should be remembered that the main purpose of the SI scale is to describe the pathogenesis of critical conditions, rather than solving specific clinical problems.

### 2.4. Phenomena of ChSI 

In chronic pathologies, similarly to in acute processes, SI as a general pathological process is not an attribute; it is a complication of many diseases with different etiologies. However, the rate of ChSI detection can characterize the severity of the nosology as a whole. A screening study of the RL values and additional ChSI phenomena in patients with various chronic diseases is presented in Table 8. The frequency of ChSI detection and a classification of cohorts based on the probability of ChSI detection are presented in Table 9. An analysis of the empirical values of SI markers in these cohorts is presented in the Appendix A. ChSI, identified by the ChSI scale, was detected at different rates in all groups of “chronic” patients (Table 8). The maximum number of ChSI cases was recorded among patients with end-stage renal disease (ESRD), systemic lupus erythematosus (SLE), and chronic limb threatening ischemia (CLI). As seen from Table 8 and Appendix A, the ChSI process was qualitatively similar in most of the groups of chronic diseases, except for ESRD and CLI, which had some signature features. Thus, the most frequent phenomenon in all groups was SIR (RL > 0), with a more than 90% probability of detection in the ESRD, SLE, and CLI groups. A semi-quantitative assessment of SIR using integral RL (Table 8) showed that most of the groups were characterized by low severity of this phenomenon (prevalence of RL-0-2), except for ESRD and CLI (prevalence of RL-1-3), and SLE, in which the dominant score was RL-3-4 and a few patients even developed RL-5. In these patients (with RL-5), the levels of proinflammatory cytokines (TNFα, IL-6, IL-8) reached values typical for hyperergic variants of septic shock and exceeded the limit of normal values by thousands and tens of thousands of times. The basic statistical characteristics of the parameters used for calculating RL in the groups of chronic diseases are given in Appendix A.

It is noteworthy (Table 8) that RL values are markedly lower in chronic pathologies than in acute critical diseases: RL-0-2 prevailed in most groups (RL-0 is absence of SIR). A notable exception was systemic lupus erythematosus (SLE), in which RL-4-5 was detected in 39% of patients. Similar SIR levels may characterize hyperergic variants of acute SI. The expression of the tissue damage phenomenon is relatively low in chronic pathologies, despite the lower threshold values of its criteria in the ChSI scale than those in the SI scale. The manifestations of adrenal dysfunction in patients with systemic autoimmune diseases are partly related to long-term hormone therapy. Similarly to in acute diseases, signs of microthrombosis (D-dimer) can characterize the presence of microcirculatory disorders regardless of other criteria.

A screening test of the ChSI scale (Table 9) showed that the cohorts studied can be differentiated into the following four clusters according to the probability of ChSI development:

1st cluster: the number of ChSI cases is rare (0–1%);

2nd cluster: detection of ChSI in up to 10% of patients;

3rd cluster: detection of ChSI in 10–50% of cases;

4th cluster: detection of ChSI in more than 50% of cases.

At the same time, diseases similar in nature, such as autoimmune diseases, can belong to all four clusters. On the other hand, a cluster may include pathologies that differ in the nature of the damaging factor. For example, the 4th cluster includes chronic limb threatening ischemia (CLI) (the main alteration factor is the inflow of tissue damage products into the bloodstream), SLE (autoimmune factors), ESRD (multiple mechanisms of alteration, including blood contact with a foreign surface) [6], APLS (chronic DIC and immunocomplex pathology). Obviously, the diseases of the 4th cluster are some of the most severe chronic pathologies. Thus, as expected, ChSI develops in inherently different diseases.

The absence of any significant correlation between the ChSI scale values and standard clinical scales of autoimmune activity, including Disease score SLEDAI for SLE (r = 0.18, *p* = 0.24) and DAS28 for RA (r = 0.29, *p* = 0.07) was also a specific feature of ChSI. Thus, the ChSI scale describes typical rather than particular pathogenesis patterns of the most severe chronic diseases and therefore does not duplicate clinical scales that are nosology specific.

### 2.5. Comparative Analysis of Acute and Chronic SI Variants

Essentially, the ChSI in most cases was similar to the time-stretched pre-SI in the acute process. This allowed the SI scale to effectively separate the obvious manifestations of acute SI from chronic pathologies (Figure 5). An exception was SLE; false-positive values of the SI scale were revealed in 26.5% of patients with this pathology. This was associated with extremely high levels of cytokinemia in this disease, which in some patients (at RL-5) could exceed the reference values of IL-6, IL-8, TNF-α by thousands and tens of thousands of times.

Analyzing the complex of SI phenomena in the combined groups (“All Chronic”/“All Acute”), we noted that the structure of the SI process in both cases had similarities (Figure 6). The frequency of the phenomena was significantly different only for DIC (*p* < 0.003, χ^2^). However, we should note that the SI scale for acute processes had higher threshold values of the criteria for neuro-endocrine distress and tissue damage/systemic alteration phenomena, while the manifestations of neuro-endocrine dysfunction in some chronic diseases were influenced by the long-term use of hormonal therapy. Taking into account these facts, as well as the differences in the intensity of RL and severity of MOD, we can conclude that the SI phenomena in acute processes were, in general, more pronounced than in ChSI. However, the principal structure of SI sub-processes in both cases was quite comparable.

### 2.6. Summary of the Results

In the “Introduction” above, we stated seven research objectives pursued in this study. Table 10 summarizes the main results of addressing these issues.

## 3. Discussion

SI as a general pathological process has both similarities of the “inflammatory mechanisms” and their realization with and fundamental differences from inflammation of the classical type (including its systemic manifestations). The pathogenetic essence of SI is a systemic “inflammatory microcirculation”, which in clinical practice manifests itself as critical (in acute processes) or latent (in chronic diseases) microcirculatory disorders.

The main patterns of SI are evident in both sepsis and acute trauma. In addition, we previously showed the determining role of SI in hospital mortality in hemorrhagic strokes [16] and the development of refractory shock in massive obstetric hemorrhage [10]. The dynamics of acute SI is characterized by the phased pattern of the process depending on the duration, clinical presentation and severity of SIR (Figure 7). The phases of SI have different degrees of criticality, which determines the need for timely prevention of the most critical phases for patients’ life: secondary phlogogenic stroke and depressive phase.

The difficulty of verifying ChSI, in turn, is associated with the lack of clear clinical equivalents and other problems:The clinical criteria for ChSI can be a sign of increasing chronic organ dysfunction. However, long-term multicenter prospective studies of a large number of diseases are needed to assess these dynamics. There is also a need to create a universal scale of chronic organ failure, similar to the SOFA scale;More accurate verification of ChSI and identification of specific stages in this process require multiple examinations of each patient over a long period of time. We have performed such studies in a limited number of patients only, so these data are not presented;It is necessary to differentiate ChSI not only from classical inflammation SIR, but also from low-grade systemic inflammation characteristic of the metabolic syndrome and type 2 diabetes mellitus. These studies are currently ongoing in our laboratory, but their results have not yet been generalized;The broader medical community has no understanding of the fundamental meaning of general pathological processes and their distinction from clinical definitions. Without such understanding, independent multicenter studies of ChSI are impossible.

The phenomena of ChSI (except for life-critical MOD) in severe chronic diseases are similar to the time-stretched state of pre-SI in intensive care patients. However, in some cases, mainly in SLE, the SIR severity in ChSI corresponds to the hyperergic phases of acute SI. We found different probabilities of ChSI occurrence in the study of the above mentioned 16 chronic diseases. This determines the possibility of subdividing the chronic pathologies into four classes. At the same time, when studying 16 diseases we could not cover the entire spectrum of nosologies potentially associated with ChSI. At the same time, the literature data show the possibility of ChSI development in severe allergic [20,21] and tumor diseases [22]. Of course, even in these cases there is a need to differentiate ChSI from systemic manifestations of classical inflammation and chronic systemic low-grade inflammation. In turn, a number of studies have shown the pathogenetic role of local and systemic inflammatory processes in the development of the tumor process, such as primary liver cancer [23,24,25,26].

Since patients with chronic conditions may have acute trauma and sepsis, there is a need to differentiate between acute and chronic SI. In general, the SI scale distinguishes clearly between these conditions, with the exception of the special, hyperergic variants characteristic of SLE. Thus, some patients in this pathology have levels of cytokinemia that are essentially fatal in acute processes. That is, the phenomenon of cytokine storm can also develop in chronic SI and is not a critical condition in this case. The question arises about the mechanisms of compensation in these patients. The mechanisms of the organism’s adaptation to the development of cytokine storm are debatable and require additional study.

The SI and ChSI scales are open systems whose components can be changed and supplemented to address specific tasks. In particular, there are well-established methods of microcirculatory disorder assessment, such as intravascular bed microscopic examination [27] and determination of blood markers of pathological endotheliocyte activation, for instance, syndecan-1 [28,29] and angiopoietin-2 [30], as well as markers of mononuclear phagocyte activation such as sTREM1 [31] and presepsin (sCD14) [32]. In addition, the evaluation of non-coding microRNAs and extracellular vesicles in the blood is a promising method for assessing systemic inflammation, which allows for the clarification of tissue sites of activation and damage in the systemic process [33,34].

The scales we have used are not a tool for solving specific clinical problems directly; rather, they define methodological approaches to their solution. Let us consider two typical examples to explain this:

1. Cytokine storm syndrome (CSS) is one of the possible manifestations of the hyperergic variant of SI determining the appropriateness of anti-cytokine therapy. The authors of [35] proposed an integral CSS verification scale (COVID-CS) adapted to COVID-19, which includes CRP, ferritin and three separate clusters of parameters: (1) albumin, lymphocytes, neutrophils, (2) D-dimers, troponin I, aminotransferases, LDH, (3) potassium, chlorine, and creatinine. The COVID-CS scale is close to the SI scale in the number and nomenclature of criteria. However, the values of this scale reflect the overall severity of the patient’s condition rather than the critical SIR levels characterizing the “cytokine storm”. At the same time, the COVID-CS scale does not contain cytokines themselves, which is probably due to the pronounced nonlinearity of changes in cytokine levels in blood. In addition, the scale does not highlight other SI sub-processes (except for SIR), and the pathological process has not been characterized comprehensively pathogenetically. At the same time, it is unclear whether this scale can be used to assess CSS in other pathologies. Therefore, to effectively address the problems of holistic assessment of pathogenesis, it is reasonable to characterize SI from the position of general pathology and then use this platform to develop clinical-diagnostic systems.

2. The universal and generally accepted SOFA scale, which we used to verify MOD, includes six (quantitative) parameters, each subdivided into four semi-quantitative ranges. One of the indicators is the platelet count. However, what does this criterion reflect in critical conditions: organ dysfunction or DIC dynamics? In this example, we see that clinical practicality prevails over theoretical validity. Of course, these two examples do not exhaust the list of methodological problems in the development of integral criteria of pathogenetic orientation in practical medicine.

The infectious variant of SI partially corresponds to the clinical definition of “sepsis” (Sepsis-3 criteria, MOD manifestations). According to our data, sepsis includes borderline conditions (pre-SI) and single cases of classical inflammation requiring intensive care. SI (predominantly the development phase and the primary phlogogenic stroke phase (cytokine storm)), in turn, can develop in individual patients before the onset of MOD criteria [10]. It is also impossible to fully associate the hyperergic variants of SI with CSS, since the critical levels of cytokinemia have not yet been officially verified (categorized) in clinical practice. The obvious clinical manifestations of any variants of acute SI are refractory shock, DIC and other signs of life-critical microcirculatory disorders.

In general, the development of SI can be defined by the term “ideal storm”: i.e., a situation arising through such a combination of a number of adverse factors that results in their total negative effect enhancing significantly and covering all of the main parameters of the system. In this case, the “system” should be understood as an integral physiological functionality of the patient’s whole organism.

## 4. Materials and Methods

### 4.1. Patients

Patients with acute critical conditions of infectious and non-infectious genesis were chosen to study acute SI, and patients with autoimmune diseases, chronic organ failure and other chronic destructive diseases were chosen to study chronic SI. A total of 26 groups were analyzed:Sepsis non-resuscitative on the 1st–2nd days—deep shin phlegmon: III-IV level of soft tissue damage in military men, all patients showing signs of MODS (average score by the SOFA scale—3.6—from 2 to 5 points). The dominant etiological factor was S. aureus. The study was conducted immediately after the surgical treatment of the inflammatory focus. Deaths and shock states in the postoperative period were not observed, treatment was carried out only in the surgical department (not in intensive care unit (ICU)), n = 40, mean age—20.4 ± 2.4 years. According to the Sepsis-3 consensus [11], this group should be attributed to the sepsis group; however, due to the difference in the clinical picture in comparison with the patients of other groups with sepsis, this group was considered separately. Signs of MOD on days 5–7 were absent in this group (MOD was resolved), so the group of patients with deep shin phlegmon on days 5–7 did not meet the inclusion criteria;Sepsis (MODS) (n = 46, mean age—47.1 ± 16.6 years, Male/Female = 60.9%/39.1%), 1–2 days after admission to the ICU, SOFA score from 2 to 10 points, mean—5.5. The initial diseases for all groups of patients with sepsis were as follows: severe pneumonia, peritonitis, obstetrical sepsis, and some other; all patients in this and in the other groups went through intensive therapy in the ICU. Lethal outcomes (n = 11) in 23.9% of cases;The same + septic shock (SS), the presence of hypotension, not responding to vasopressors, SOFA score from 6 to 14 points, mean—9.75 (n = 14, mean age—49.1 ± 17.8 years, Male/Female = 57.1%/42.9%). Lethal outcomes (n = 10) in 71.4% of cases;Sepsis (MODS) (n = 13, mean age—40.2 ± 14.2 years, Male/Female = 61.5%/38.5%) screening on day 5–7 of hospitalization in the ICU, SOFA score from 3 to 10, mean—5.7. Lethal outcomes (n = 4) in 30.7% of cases;Tertiary peritonitis (TP) with MODS, and prolonged (14–30 days) and subacute (>30 days from the start of manifestation) septic process (n = 34, mean age—51.5 ± 16.6 years, Male/Female = 58.8%/41.2%), SOFA score was not calculated. Lethal outcomes (n = 10) in 29.4% of cases;The same + development of septic shock (TP + SS) (n = 17, mean age—50.2 ± 15.6 years. Male/Female = 64.7%/35.3%), SOFA score was not calculated. Lethal outcomes (n = 16) in 94.1% of cases;Polytrauma-acute multiple injuries in two or more different body regions, required intensive therapy in the ICU (n = 51, mean age—37.8 ± 14.9 years, Male/Female = 67.4%/32.6%), on the 1st–2nd days of hospitalization in the ICU, with the development of MODS, SOFA score from 2 to 12, mean—4.96. Lethal outcomes (n = 11) in 21.6% of cases. All patients had values of the ISS scale ≥ 9. According to the ISS criteria, the following injuries prevailed: chest in twenty-three patients; extremities, pelvic organs and abdomen in twenty patients; craniocerebral trauma in five patients; and mixed variants in three patients. All patients with predominant chest injuries had pulmonary insufficiency-ALI/ARDS. ROC analysis was used exclusively to investigate the integral ALI/ARDS group, in which 14 out of 37 patients had signs of pulmonary failure without MOD (Table 8). In all other patients with acute pathologies in this study, the presence of MOD was established according to the criteria of the SOFA scale. ISS scale values were determined on the 1st day of hospitalization, and SOFA and CI scales were scored in dynamics. In patients with chest trauma, the APACHE II and TRISS scales were additionally used for scoring on the first day of follow-up. In other cases, these scales were applied to only some of the patients;Multiple injuries with the development of MODS (n = 18, mean age—39.4 ± 15.1 years, Male/Female = 64.7%/35.3%), on days 5–7 after admission to the ICU, SOFA score from 2 to 16, mean—6.75. Traumatic lesions in the following areas: thorax, upper and lower extremities, pelvic region, abdominal injuries, and traumatic brain injury. Lethal outcomes (n = 9) in 50.0% of cases.

The 28-day mortality in the entire sample of acute nosologies was 71 cases.

The lethality was determined at 1–10 days under acute sepsis and trauma and up to 39 days under subacute sepsis. There were 71 deaths in total, and in all cases the clinical diagnoses were confirmed by autopsy. In all cases, autopsy found microcirculatory disorders: edema of internal organs and brain membranes, the presence of microthrombosis and microbleeds, atrophic changes in the myocardium, liver and kidneys, and in some cases the atrophy or necrosis of adrenal glands, noncardiogenic pulmonary edema, and brain intrusion.

The exclusion criteria for the polytrauma groups: presence of severe chronic pathology, autoimmune and auto-inflammatory diseases, sepsis, diabetes, and other nosologies. The exclusion criteria for all groups with acute conditions: presence of major arterial thrombosis, strokes, heart attacks, and pulmonary embolism in the acute period. Anti-cytokine therapy was an exclusion criterion for all groups, including chronic diseases (see below);

9.Systemic lupus erythematosus—SLE (n = 49, mean age—43.7 ± 13.3 years, Male/Female = 6.1%/93.9%). The patients met the 1982 American College of Rheumatology criteria for SLE [36]. The duration of SLE was (mean ± SD) 11.9 ± 9.4 years. Disease activity was assessed according to the SLE Disease Activity Index (SLEDAI) [37]. SLE patients with SLEDAI ≥ 5 (95.5% of cases) were attributed to the active-disease cohort, and those with SLEDAI < 5 (4.5% of cases) were attributed to the stable-disease cohort;10.Rheumatoid arthritis—RA (n = 42, mean age—53.1 ± 14.3 years, Male/Female = 9.5%/90.5%). The patients met the 1987 American College of Rheumatology criteria for RA [38]. Thirty-eight patients (90.5%) were seropositive. RA activity was measured by Disease Activity Score in 28 joints (DAS28) [39]: low activity (DAS28 ≤ 3.2) was detected in 7.2% of cases, moderate activity (3.2 < DAS28 ≤ 5.1) in 35.7%, and high activity (DAS28 > 5.1) in 57.1%. The duration of RA was (mean ± SD) 7.1 ± 7.4 years;11.Reactive arthritis associated with *Chlamydia trachomatis*—ReA (n = 30, mean age—42.4 ± 14.3 years, Male/Female = 39.3%/60.7%). The duration of ReA was (mean ± SD) 7.1 ± 7.4 years;12.Ankylosing spondylitis—AS (n = 27, mean age—41.0 ± 13.1 years, Male/Female = 85.2%/14.8%). The patients met the modified New York criteria for AS (1984) [40]. AS activity was measured by Bath Ankylosing spondylitis Disease Activity Index—BASDAI (1994) [41], the median of AS activity was 4.05. High disease activity (BASDAI > 4) was detected in 73% of cases;13.Psoriatic arthritis—PsA (n = 12, mean age—52.9 ± 6.1 years, Male/Female = 50%/50%). The patients met the CASPAR (for Classification Criteria for Psoriatic Arthritis) criteria [42];14.Chronic rheumatic valvular heart disease—RHD (n = 15, mean age—55.3 ± 13.0 years, Male/Female = 14.3%/85.7%). The diagnosis of RHD was based on echocardiographic findings according to the World Heart Federation criteria [43];15.Chronic heart failure—CHF (n = 49, mean age—80.7 ± 4.3 years, Male/Female = 73.5%/26.5%). Patients included in the study were 70 years old or above, diagnosed with CHF, taking medications, having heart failure of New York Heart Association (NYHA) functional class II-IV, and were clinically stable. Most patients had chronic bronchitis, pulmonary emphysema, age-related encephalopathy. The exclusion criteria were autoimmune diseases, tumors, end-stage renal failure, previous stroke with immobility, severe dementia, and acute inflammatory processes at the time of the study;16.End-stage renal disease—ESRD (n = 42, mean age—45.4 ± 13.9 years, Male/Female =47.6%/52.4%) caused by chronic glomerulonephritis (n = 22), chronic pyelonephritis (n = 12) and diabetic nephropathy (n = 8). The diagnosis of ESRD was based on the international criteria K/DOQI (2002) [44]. All patients received replacement treatment by programmed hemodialysis-12 h a week. The duration of the dialysis period was 63.0 ± 9.6 months. Blood samples were collected before hemodialysis sessions;17.Chronic limb threatening ischemia—CLI—caused by common femoral artery atherosclerotic lesions (n = 38, mean age—65.8 ± 9.1 years, Male/Female = 67.6%/32.4%). All patients had grade III CLI according to Rutherford classification [45]. The study was conducted in preparation for a high thigh amputation. The condition of the patients was assessed as moderate in 68.2% of cases and as severe in 31.8%;18.Chronic nonhealing wounds—CNW in military men (n = 42, mean age—19.4 ± 0.5 years) caused by surgical treatment of phlegmon (50.5% of cases), microtraumas and abrasions (44.5% of cases), and erysipelatous inflammation (5% of cases). The duration of the purulent process in all patients was more than 90 days. The depth of soft tissue lesion corresponded to grade II (47.5% of cases), grade III (31.5% of cases), and grade IV (21% of cases) by D. Ahrenholz classification (1991), length: from 10 to 25 cm^2^;19.Autoimmune thyroiditis—AIT (n = 29, mean age—44.2 ± 13.2 years). Thyroid gland hypofunction was detected in twenty-one patients and hyperfunction in eight patients;20.Pelvic inflammatory disease—PID (n = 16, mean age—28.9 ± 4.7 years)—chronic infection of the upper genital tract. Patients were treated for recurrent first trimester miscarriage;21.Menopausal syndrome: women who were treated for menopausal syndrome (n = 16, mean age—50.7 ± 4.4 years). All patients had stage 2 arterial hypertension (according to the criteria of the American College of Cardiology/American Heart Association [46]);22.Chronic renal allograft dysfunction—CRAD (n = 23, mean age—42.0 ± 9.4 years Male/Female = 69.6%/30.4%): patients after renal allotransplantation with morphological (according to Banff classification [47]), clinical, and laboratory signs of CRAD (sustained increase in plasma creatinine over 0.15 mmol/L, persistent proteinuria over 0.5 g/day, arterial hypertension). Signs of allograft dysfunction manifested themselves in all recipients not earlier than 6 months after transplantation. All patients received hemodialysis therapy before renal transplantation;23.Normal function of renal allograft: patients with normal renal allograft function, i.e., without clinical and laboratory signs of CRAD (n = 24, mean age—43.5 ± 9.1 years Male/Female = 50%/50%). It was a reference group for the CRAD group;24.Primary antiphospholipid syndrome (PAPS) in women with recurrent miscarriage (n = 5). All patients met the PAPS classification criteria [48];25.Control group—healthy blood donors aged 18–55 years (n = 50, mean age—34.1 ± 10.4 years, Male/Female = 52%/48%) recruited at Regional Blood Transfusion Station (Ekaterinburg);26.Elderly people aged 60–74 years without acute and system destructive diseases, acute attack of chronic diseases, and free from other inflammatory conditions (n = 22, mean age—68.5 ± 5.9 years, Male/Female = 59.1%/40.9%), which underwent a routine medical check-up at Ekaterinburg City Clinical Hospital No. 40.

The research protocol was approved by the Ethical Committee of the IIP UB of RAS. All subjects who participated in this research provided written informed consent.

### 4.2. Measurement of Biomarkers

We studied a blood plasma stabilized by citrate, previously frozen at −20 °C. The levels of interleukins-6,8,10, TNFα, CRP, cortisol, myoglobin, troponin I, and D-dimers in blood plasma samples were analyzed using a closed system for immunochemiluminometric assay, Immulite (Siemens Medical Solutions Diagnostics, Malvern, PA, USA).

### 4.3. Verification of Systemic Inflammation

We used SI and ChSI Scales to identify the SI (ChSI), its phases, and complex phenomena structure, according to the above-described method (Table 2, Table 3 and Table 4).

### 4.4. Statistical Analysis

Statistical analyses were performed using SPSS for Windows 15.0 (SPSS Inc., Chicago, IL, USA) and Statistica 12.0 program (Stat Soft, Inc., Tulsa, OK, USA). The descriptive statistics are presented by their main characteristics: m—mean value, Me—median, SD—standard deviation, 25% ÷ 75%—quartiles, and Minimum-Maximum. Kolmogorov-Smirnov and Shapiro-Wilk tests were used to check the hypothesis that the distribution of samples was not normal. We used Spearman correlation coefficient (r), a non-parametric analytical method. Comparisons between the groups were performed using the Chi-square (χ^2^) test for categorical variables. The areas under the receiver operating characteristic curve (AUC) were used to evaluate the ability of the biomarkers, RL, and SI scale to discriminate survivors from non-survivors. The ratio of the AUC and diagnostic significance intervals were evaluated as: 0.9–1.0—“excellent”, 0.8–0.9—“very good”, 0.7–0.8—“good”, 0.6–0.7—“medium”, 0.5–0.6—“unsatisfactory” [19]. Differences in AUCs were determined using the Medcalc for Windows 12.2.1.0 (MedCalc Software Ltd., West-Vlaanderen, Belgium). All of the results were considered statistically significant if the *p*-value was <0.05. The diagrams were created using Microsoft Office Excel 2007.

## 5. Conclusions

Lethality in intensive care units is associated with the progression of MOD up to the development of refractory shock. In most cases, SI is the common pathogenetic platform for these events, regardless of the nature of the damaging factor (infectious and non-infectious). In particular, in sepsis, relatively moderate MOD (mean SOFA score-3.6 in the non-resuscitative sepsis-3 group) without any significant signs of SI does not lead to mortality, while rapidly progressing MOD and especially septic shock in intensive care patients, are organically associated with the development of critical phases of SI, where high mortality is likely. In this case, it is reasonable to consider SI not as a specific clinical definition, but as a common pathological process, which is not identical to SIR in classical inflammation. Moreover, on this theoretical and methodological basis it is reasonable to form models of clinical definitions, first of all, of resuscitation syndromes, solving specific tasks of medical practice. Thus, the criticality of ICU patients’ conditions is determined by pathogenetic association of MOD and SI. At the same time, the initial signs of SI may precede the clinics of critical condition, which allows for the use of SI criteria for the prognosis of the development of these conditions. The chronic variant of SI currently has no clear clinical analogues, and the role of this process needs a focused and comprehensive study, including long-term prospective studies.

## Figures and Tables

**Figure 1 ijms-24-01144-f001:**
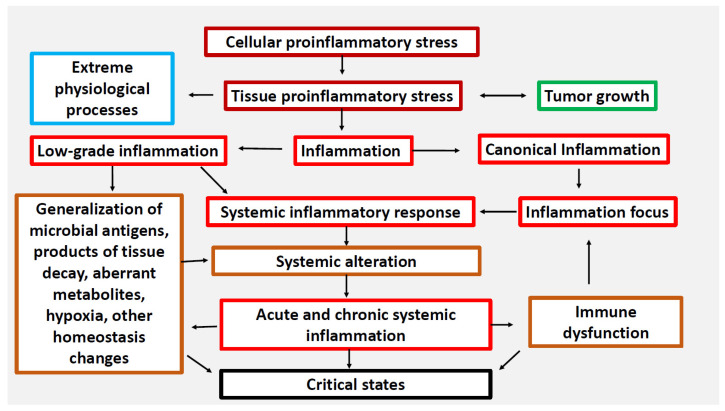
Relationship between cellular and tissue stress and human pathology [2]. Note. Cellular proinflammatory stress is a typical cell response to various real or potential damage of macromolecules as well as to external stressors. It is a common basis for various pathological and some physiological processes, including muscle contraction, tissue growth, mucosal function, and the exercise of normal lymphocyte selection in primary lymphoid organs. Tumor cells and tumor microenvironment cells are under stress, but tumor growth is not inflammation. Systemic hyperinflammation is a dysfunctional process, a complication of other pathological processes that initiate systemic alteration.

**Figure 2 ijms-24-01144-f002:**
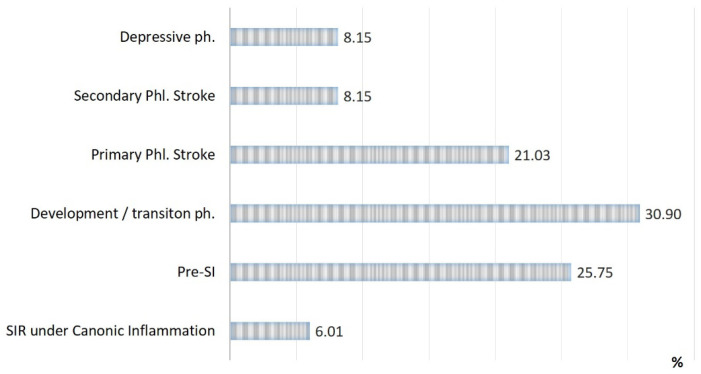
Distribution of patients in the study groups by type of inflammation and phases of SI (n = 233).

**Figure 3 ijms-24-01144-f003:**
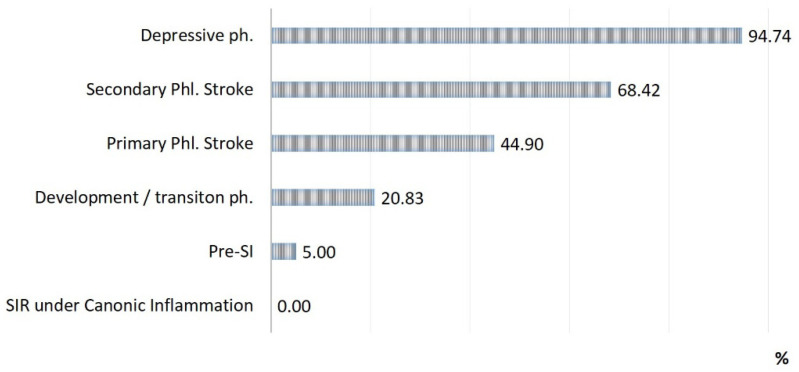
Lethality in each phase of SI (n = 71).

**Figure 4 ijms-24-01144-f004:**
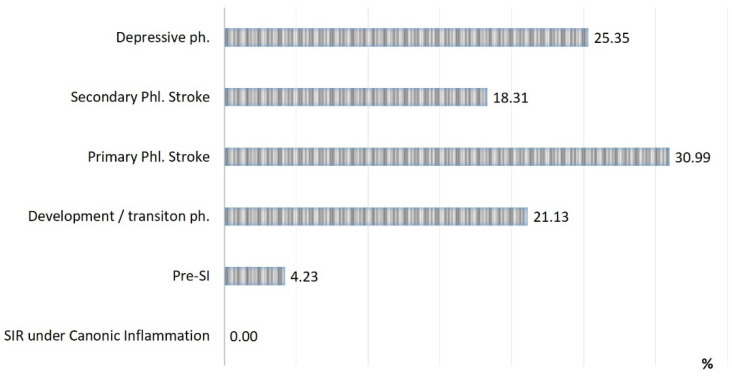
Distribution of all lethal cases by SI phases (n = 71).

**Figure 5 ijms-24-01144-f005:**
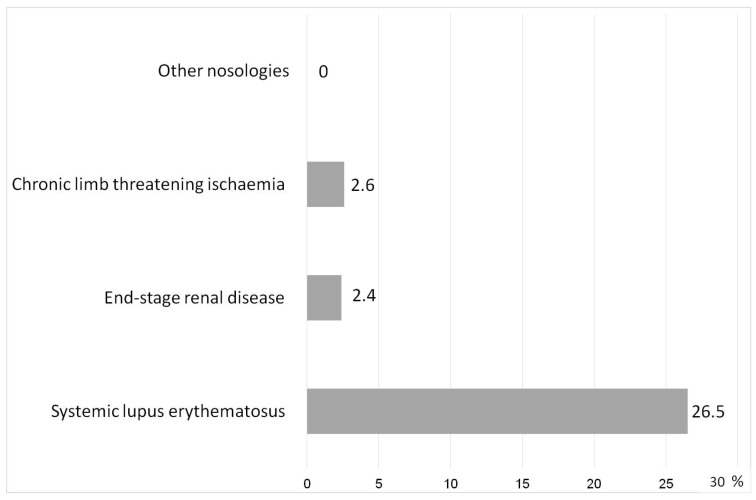
Rate of false positive values by the SI scale in chronic diseases (%).

**Figure 6 ijms-24-01144-f006:**
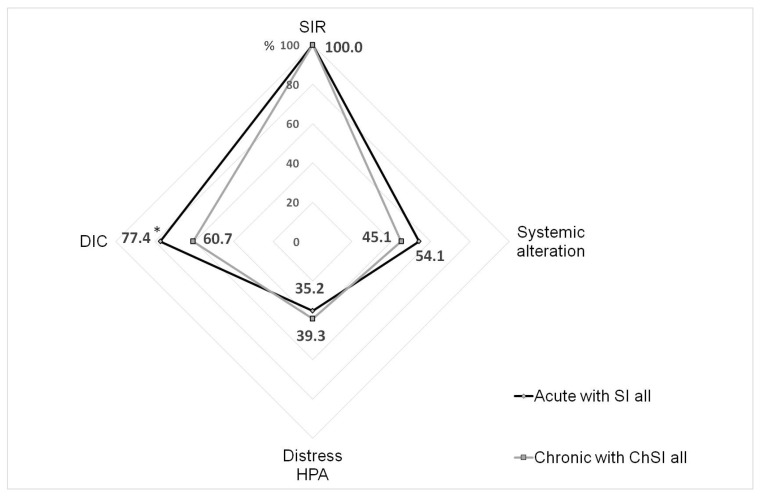
The frequency of the phenomena detected in the integral groups of patients with acute and chronic SI. Note. *—significant difference (Chi-square test, *p* < 0.05); SIR—systemic inflammatory response (RL > 0); DIC—disseminated intravascular coagulation; HPA—hypothalamic-pituitary-adrenal system; SI—systemic inflammation; ChSI—chronic systemic inflammation.

**Figure 7 ijms-24-01144-f007:**
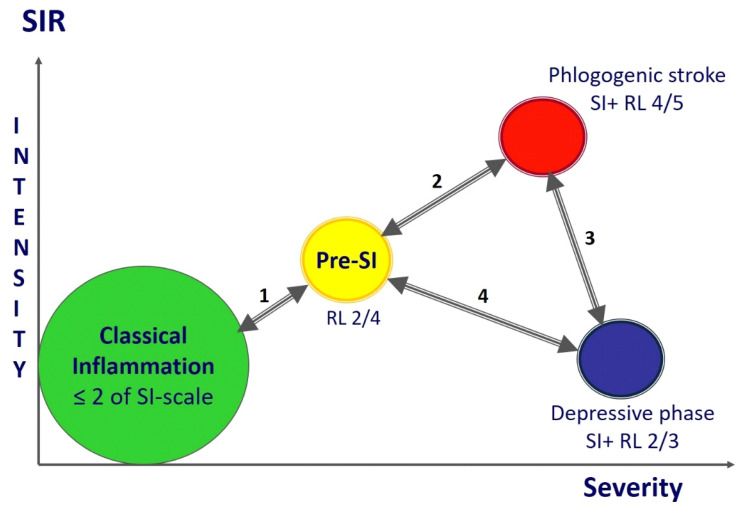
SI phases («punching» variant). Model scheme of the SI dynamics with variants of phases and interphase transitions during the gradual development of the process. Note. 1—Reversal from the pre-SI or SI development phase to the systemic manifestations of classical inflammation is the most favorable variant of the dynamics that is associated with SI relief and resolution (the SI development phase is not marked in the figure, but is in the same area as pre-SI); 2—in principle, the process of transition to hyperergic state is reversible in the early stages, but the process can be complicated by refractory shock and a complex of other life-threatening syndromes leading to lethal outcome; 3—the unfavorable dynamics variant regardless of the reversibility of hyper- and hypoergic critical SI phases, especially during the formation of the secondary phlogogenic stroke phase (5–8 days); 4—the transition to the pre-SI or resolution phase (reversible development) from the stable depressive phase of SI is unlikely, especially on in the longer term (>14 days) in the development of the critical condition.

**Table 1 ijms-24-01144-t001:** The ranges of biomarker concentrations and their corresponding RLs.

Marker	Normal	Ranges of Absolute Parameter Values and Corresponding RL					
		1	2	3	4	5	6
IL-6, pg/mL	<5.0	5–10	10–40	40–200	200–1000	>1000	-
IL-8, pg/mL	<10.0	10–25	25–100	100–500	500–2500	>2500	-
IL-10, pg/mL	<5.0	-	5–10	10–25	25–100	100–500	>500
TNFα, pg/mL	<8.0	8–16	16–40	40–160	160–800	>800	-
CRP, mg/dL	<1.0	1–3	3–15	>15	-	-	-

Note. The mean of three maximal RL values for each factor gave the RL (1 to 5) for each patient, which was used for further calculation of the SI score. RL—reactivity level; IL—interleukin; TNF—tumor necrosis factor; CRP—C-reactive protein. It is reasonable to determine the RL by the degree (multiplicity) to which it exceeds the maximum permissible values of the norm. For the method we use, the normal limits are given in the Normal column, and the ranges for determining RL are given in absolute values (pg/mL and mg/mL).

**Table 2 ijms-24-01144-t002:** Integral SI scale.

Phenomena	Criteria	Points	Note
SIR–Cytokinemia	Values of RL-0-5	2–5	Values of RL-0-1 except for acute SI
DIC	D-dimer > 500 ng/mL	1	or DIC-syndrome, e.g., DIC scale ≥ 5 score
Distress of hypothalamic-pituitary-adrenal axis	Cortisol > 1380 or <100 nmol/L (Norm 170–690 nmol/L)	1	In the absence of the criteria; but for glucocorticoid therapy ^1^, +1 point to score
Systemic alteration	Troponin I ≥ 0.2 ng/mL and/or myoglobin ≥ 800 ng/mL ^2^	1	Troponin does not sum up in case of myocardial infarction
MOD	SOFA score and/or criteria of MODS	1	Phenomenon and syndrome are non-specific to SI

Note. Calculation principle: each detected phenomenon is assigned a certain number of points, and then the points are summed up. A total SI score 3–4 indicates the development of pre-SI and a score > 5 indicates the development of SI in the patient; ^1^—glucocorticoid therapy for more than 1 day and no less than 100 mg/day (prednisolone), ^2^—criterion of myoglobin >800 ng/mL is used in case of local injury of muscular tissue and >60 ng/mL without local injury. SI—systemic inflammation; SIR—systemic inflammatory response; DIC—disseminated intravascular coagulation; MOD—multiple organ dysfunction; RL—reactivity level; SOFA—sequential organ failure assessment score; MODS—multiple organ dysfunction syndrome.

**Table 3 ijms-24-01144-t003:** Verification of SI phases using SI and RL integral scales.

Verification	Scale-RL (Scores)	Scale-SI (Scores)
SIR without SI and pre-SI	1–2	≤2
Pre-SI	1–4	3–4
SI	2–5	5–9
Phases (SI) development/permission	2–3	5–7
Phlogogenic stroke (PS)	4–5 *	5–9 *
Depressive phase (DP) (depletion phase)	2–3 **	5–7 **
The structure of the SI process complex	Ratio of individual SI phenomena	

Note. *—in acute processes, we determine the development of primary phlogogenic stroke phase (PPS, cytokine storm) early in the observation period (days 1–3), and—of secondary phase (SPS) in the longer term; **—in the presence of shock and/or resuscitation syndromes, such as DIC (consumption stage), ARDS (secondary acute respiratory distress syndrome), taking into account the time of their onset. SI—systemic inflammation; SIR—systemic inflammatory response; RL—reactivity level.

**Table 4 ijms-24-01144-t004:** The Chronic Systemic Inflammation (ChSI) scale.

ChSI Phenomena	Partial ChSI Criteria	Norm	ChSI Scale Points
Systemic inflammatory response	RL scale (Points) (0 to 5)	0	1 RL point = 1 ChSI scale point
Microthrombosis	D-dimers > 500 ng/mL	<250	1 point
Systemic alteration	Myoglobin > 60 ng/mL or	<25	1 point
	Troponin I > 0.2 ng/mL	<0.2	
Distress of the hypothalamic-pituitary-adrenal system	Cortisol > 690 nmol/L or	138–690	1 point
	Cortisol < 100 nmol/L		

Note. Calculation principle: each detected phenomenon is assigned a certain number of points, and then the points are summed up. A total ChSI score ≥ 3 indicates the development of ChSI in a patient.

**Table 5 ijms-24-01144-t005:** Frequency of different RL and SI phenomena values in the acute nosology groups studied (%).

Group	RL, %						SIR	Systemic Alteration	DIC	Distress HPA
	0	1	2	3	4	5				
Control 1 (donors), n = 50	100	0	0	0	0	0	0	0	0	0
Control 2 (elderly people), n = 22	81.8	18.2	0	0	0	0	18.2 ^1^	0	0	0
Sepsis non-resuscitative on days 1–2, n = 40	0	27.5	55.0	17.5	0	0	100.0	5.0	10.0	0
Sepsis on days 1–2, n = 46	0	4.4	10.9	41.3	30.4	13.0	100.0	34.8	58.7	30.4
Sepsis on days 5–7, n = 13	0	0	7.7	46.2	46.1	0	100.0	46.2	92.3 ^2,3^	15.4
Septic shock, n = 14	0	0	7.1	14.3	42.9	35.7	100.0	42.9	85.7	64.3
Tertiary peritonitis, n = 34	0	0	14.7	64.7	17.7	2.9	100.0	32.4	85.3	5.9
Tertiary peritonitis + SS, n = 17	0	0	35.3	58.8	5.9	0	100.0	70.8	88.2	35.3
Multiple injuries on days 1–2, n = 51	0	2.0	25.5	37.2	31.4	3.9	100.0	52.9	72.6	31.4
Multiple injuries on days 5–7, n = 18	0	5.6	44.4	22.2	27.8	0	100.0	50.0	55.6 ^3^	50.0

Note. RL—reactivity level; SI—systemic inflammation; SIR—systemic inflammatory response; DIC—disseminated intravascular coagulation; HPA—hypothalamic-pituitary-adrenal axis; SS—septic shock. ^1^—manifestations of SIR in this category of patients can also be associated with chronic low-intensity inflammation connected with age-related changes in homeostasis (allostasis). ^2^—statistically significant differences in the same nosology group on different days of the study (Chi-square test, *p* < 0.05); ^3^—statistically significant differences between the groups “Sepsis 5–7” and “Multiple injuries 5–7” (Chi-square test, *p* < 0.05).

**Table 6 ijms-24-01144-t006:** The rate of phases of acute SI, registration of SI, and lethal outcomes in the MOD groups (%).

Group (n)	Pre-SI, %	Phases of SI (100%—All SI Cases)			SI, %	LO, %
		Development/Interphase Transition/Permission	Phlogogenic Stroke ^1^	Depressive		
Control 1 (donors), n = 50	0	0	0	0	0	0
Control 2 (elderly people), n = 22	0	0	0	0	0	0
Sepsis non-resuscitative on days 1–2, n = 40	65.0	7.5	0	0	7.5	0
Sepsis on days 1–2, n = 46	21.7	41.2	58.8	0	73.9 ^3^	23.9
Sepsis on days 5–7, n = 13	0	46.2	53.9	0	100 ^3,4^	30.8
Septic shock, n = 14	0	0	78.6	21.4	100	71.4
Tertiary peritonitis, n = 34	17.6	75.0	25.0 ^2^	0	82.4	29.4
Tertiary peritonitis + SS, n = 17	0	0	5.9 ^2^	94.1	100	94.1
Multiple injuries on days 1–2, n = 51	21.6	53.3	46.7	0	76.5	21.6
Multiple injuries on days 5–7, n = 18	44.4	50.0	50.0 ^2^	0	55.6 ^4^	50.0

Note. SI—systemic inflammation; MOD—multiple organ dysfunction; SS—septic shock; LO—lethal outcomes. ^1^—primary phlogogenic stroke phase (cytokine storm) was determined on the 1st–2nd days of observation; secondary phlogogenic stroke, on the 5th–7th days and thereafter; ^2^—secondary phlogogenic stroke phase. ^3^—statistically significant differences in the same nosology group on different days of the study (Chi-square test, *p* < 0.05); ^4^—statistically significant differences between the groups “Sepsis 5–7” and “Multiple injuries 5–7” (Chi-square test, *p* < 0.05).

**Table 7 ijms-24-01144-t007:** Area under the ROC curve (AUC ± standard error) for the studied scales in the prognosis of fatal outcomes in patients (n-37) with predominantly chest injury.

Scales	Overall Mortality Prognostic Value (AUC) on Days 1–10 after Injury (n-9). Scale Scored on Day 1.	Overall Mortality Prognostic Value (AUC) on 6–10 Days (n-7). Scales Scored on 5–7 Day.
SI scale	**0.862 ± 0.0514 (*p* < 0.0001)**	**0.874 ± 0.0567 (*p* < 0.0001)**
RL scale	**0.867 ± 0.051 (*p* < 0.0001)**	**0.878 ± 0.0574 (*p* < 0.0001)**
SOFA	**0.721 ± 0.0792 (*p* = 0.0052)**	**0.68 ± 0.0867 (*p* = 0.0376)**
GCS	**0.821 ± 0.0722 (*p* < 0.0001)**	**0.801 ± 0.0771 (*p* = 0.0001)**
SIRS	**0.748 ± 0.068 (*p* = 0.0003)**	**0.742 ± 0.072 (*p* = 0.0008)**
APACHE II	0.539 ± 0.153 (*p* = 0.7995)	We did not.
TRISS	**0.77 ± 0.10 (*p* = 0.0052)**	We did not.
ISS	**0.78 ± 0.11 (*p* = 0.008)**	We did not.
TISS	0.625 ± 0.2 (*p* = 0.5329)	We did not.
SAPS II	0.537 ± 0.211 (*p* = 0.8613)	We did not.

Note. APACHE II, TRISS, ISS, TISS, SAPS II scales according to the clinical protocol were defined only on the 1st day of hospitalization. The diagnostic significance of the scales according to AUC values [19]: 0.6–0.7 (Medium), 0.7–0.8 (Good), 0.8–0.9 (Very good), 0.9–1.0 (Excellent). AUC—area under the ROC curve; RL—reactivity level; SI—systemic inflammation; SOFA—sequential organ failure assessment score; GCS—Glasgow coma scale; SIRS—Systemic Inflammatory Response Syndrome; APACHE II-acute physiology and chronic health evaluation; TRISS-trauma and injury severity score; ISS—injury severity score; TISS—therapeutic intervention scoring system; SAPS II—simplified acute physiology score. Statistically significant results are highlighted in bold.

**Table 8 ijms-24-01144-t008:** Frequency of RL values and basic ChSI phenomena in groups of patients with chronic nosologies.

Group	RL, %						SIR, %	Systemic Alteration %	Microthrombosis, %	Distress of HPA, %
	0	1	2	3	4	5				
Control 1 (donors), n = 50	100	0	0	0	0	0	0	0	0	0
Control 2 (elderly people), n = 22	81.8	18.2	0	0	0	0	18.2	0	0	0
SLE, n = 49	8.2	4.1	16.3	32.6	34.7	4.1	91.8	6.1	40.8	28.6
RA, n = 42	31.0	47.6	19.0	2.4	0	0	69.0	7.1	54.8	28.6
ReA, n = 30	46.7	33.3	20.0	0	0	0	53.3	0	23.3	43.3
AS, n = 27	44.5	33.3	22.2	0	0	0	55.5	3.7	11.1	7.4
PsA, n = 12	33.3	50.0	16.7	0	0	0	66.7	8.3	8.3	8.3
RHD, n = 15	53.5	33.3	13.2	0	0	0	46.7	0	13.3	14.3
CHF, n = 49	53.1	36.7	10.2	0	0	0	46.9	2.0	33.3	2.0
ESRD, n = 42	4.8	16.6	54.8	21.4	2.4	0	95.2	92.9	38.1	14.3
CLI, n = 38	5.3	31.6	52.6	10.5	0	0	94.7	50.0	47.4	34.2
CNW, n = 42	19.0	78.6	2.4	0	0	0	81.0	2.4	9.5	45.2
AIT, n = 29	79.3	20.7	0	0	0	0	20.7	0	0	3.4
PID, n = 16	75.0	25.0	0	0	0	0	25.0	- *	0	- *
Menopausal syndrome, n = 16	93.7	6.3	0	0	0	0	6.3	0	0	0
CRAD, n = 23	8.7	69.6	17.4	4.3	0	0	91.3	30.4	21.7	56.5
Normal function of renal allograft, n = 24	58.3	25.0	16.7	0	0	0	41.7	0	4.2	29.2
PAPS, n = 5	0	0	20.0	80.0	0	0	100	- *	100	- *

Note. RL—reactivity level; SI—systemic inflammation; ChSI—chronic systemic inflammation; SLE—systemic lupus erythematosus; RA—rheumatoid arthritis; ReA—reactive arthritis; AS—ankylosing spondylitis; PsA—psoriatic arthritis; RHD—chronic rheumatic valvular heart disease; CHF—chronic heart failure; ESRD—end-stage renal disease; CLI—chronic limb threatening ischemia; CNW—chronic nonhealing wounds; AIT—autoimmune thyroiditis; PID—pelvic inflammatory disease; CRAD—chronic renal allograft dysfunction; PAPS-primary antiphospholipid syndrome. *—markers of the phenomenon were not measured in this group.

**Table 9 ijms-24-01144-t009:** The classification of chronic diseases based on the probability of ChSI signs detection.

Disease Clusters	Nosologies (The Frequency of ChSI Detection, %)
Cluster 1 (≤1% of ChSI)	Autoimmune thyroiditis (0 %) ^1^
	Menopausal syndrome (0 %) ^1^
	Pelvic inflammatory disease (0 %) ^1^
	Normal function of renal allograft (0 %) ^1^
Cluster 2 (1.1–10% of ChSI)	Chronic heart failure (2.0%)
	Chronic nonhealing wounds (9.5%)
	Psoriatic arthritis (8.3%)
Cluster 3 (10.1–50% of ChSI)	Ankylosing spondylitis (11.1%)
	Chronic rheumatic valvular heart disease (13.3%)
	Reactive arthritis (20.0%)
	Rheumatoid arthritis (31.0%)
	Chronic renal allograft dysfunction (43.5%)
Cluster 4 (>50% of ChSI)	Chronic limb threatening ischemia (57.9%)
	Systemic lupus erythematosus (75.5%)
	End-stage renal disease (88.1%)
	Primary antiphospholipid syndrome (100%)

Note. ^1^—early results.

**Table 10 ijms-24-01144-t010:** The main outcomes of the study.

Objective	Solution	Result
1. Check the quality of the SI scale.	Comparison of the SI scale score with the clinical equivalents of SI.	The SI scale detected SI in 100% of cases of refractory shock, in 96% of fatal outcomes, and in only 7.5% of non-resuscitated sepsis (MOD) cases.
2. Confirm the general pathological nature of SI.	Comparison of two resuscitation cohorts: with sepsis and acute trauma.	Principal similarities in SI development revealed among pathologies with different nature of systemic damage
3. Verify the individual SI phases.	Comparison of hyperergic and hypoergic variants of SI at different terms of critical state development.	Five principal phases of SI identified, of which the phase of secondary phlogogenic stroke (cytokine storm) and the depressive phase were the most critical for patients’ life (lethality > 50%). They were detected only in intensive care patients.
4. Analyze the mortality prognostic value of the SI and clinical scales in polytrauma.	Comparison of the prognostic effectiveness of SI and RL with the clinical scales: APACHE II, TRISS, ISS, GCS, SOFA, TISS, SAPS II, SIRS.	The SI scales were superior or, at least, not inferior to the clinical scales tested in prognostic value for fatal outcomes.
5. Define the role of SI in chronic diseases.	Identification of signs of ChSI in 16 chronic pathologies with alteration factors of different nature.	The diseases were divided into four classes according to the degree of ChSI detection. Assignment to each class was determined by the intensity of systemic damage rather than its nature.
6 Differentiate between acute and chronic SI.	Determination of false-positive SI scale results in chronic diseases	False-positive results recorded in significant numbers (26.5%) only in patients with SLE who had very high levels of cytokinemia.
7. Address the identification of similarities and differences between acute and chronic SI.	Comparison of SI phenomena frequency in integral groups with acute and chronic pathology	In general, the severity of SI phenomena in acute pathologies was higher, but the basic structure of their ratios was comparable.

## Data Availability

The datasets analyzed during the current study are available from the corresponding author on reasonable request as they contain information on the gender, age, and diagnosis of the patients.

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
