# Peer review of "Acute and Chronic Systemic Inflammation: Features and Differences in the Pathogenesis, and Integral Criteria for Verification and Differentiation"

_ijms, 2023, doi:10.3390/ijms24021144_

Round 1

Reviewer 1 Report (New Reviewer)

The authors present an interesting manuscript which aims to identify the difference between acute and chronic infection.

The work is well written and structured. The following parameters are analysed: interleukins 6, 8, 10; tumor necrosis factor alpha; C-reactive protein; D-dimer; cortisol; troponin I; myoglobin.

Why did the authors exclude sTREM1 and procalcitonin?

I recommend citing this manuscript DOI: 10.1080/10520295.2018.1535138.

Furthermore, there are also currently new molecular biology methods such as miRNAs. I recommend inserting a paragraph justifying the decision not to use the mirna.

it might be helpful to cite this article:

The authors should envisage a subsequent extension of the study aimed at identifying miRNAs expressed in chronic and acute inflammation DOI: 10.3390/ijms23169354

Author Response

Dear Reviewer,

Thank you very much for your kind response with your valuable advice. We enclose our corrected manuscript with all highlighted changes. Next, we respond to all your comments point by point.

The authors present an interesting manuscript which aims to identify the difference between acute and chronic infection.

The work is well written and structured. The following parameters are analysed: interleukins 6, 8, 10; tumor necrosis factor alpha; C-reactive protein; D-dimer; cortisol; troponin I; myoglobin.

Why did the authors exclude sTREM1 and procalcitonin?

I recommend citing this manuscript DOI: 10.1080/10520295.2018.1535138.

We described the criteria for the selection of markers included in this study in section 3.4. Initially, we selected more than 200 potential markers for the verification of systemic inflammation (including sepsis), including TREM1 and procalcitonin. In this case, sTREM1, as well as the functionally similar marker sCD14 (presepsin), can be considered as an indicator mainly reflecting monocyte/macrophage function. In our studies, we used other markers of phagocyte activation (eosinophil cationic protein and CD64 in the blood). These data are not presented in current study because these indices were determined only in some groups. In the "Discussion" section, we additionally noted the appropriateness of sTREM1 and presepsin measuring in acute and chronic variants of systemic inflammation and added appropriate references (lines 615-616).

Procalcitonin is used to assess the patient condition in acute processes, mainly in sepsis. In one of our works we also described its increase in acute aseptic process (stroke) and presented a modernized SI scale, which included procalcitonin [doi: 10.17116/jnevro202012008224]. However, procalcitonin is not a universal indicator that can characterize both acute and chronic systemic inflammation with equal efficiency. That is why we did not include procalcitonin in the integral scales of acute and chronic SI.

Furthermore, there are also currently new molecular biology methods such as miRNAs. I recommend inserting a paragraph justifying the decision not to use the miRNA.

It might be helpful to cite this article:

The authors should envisage a subsequent extension of the study aimed at identifying miRNAs expressed in chronic and acute inflammation DOI: 10.3390/ijms23169354

We agree with your comment about the importance of miRNA in the development of systemic inflammation, including sepsis. Therefore, we have further described the role of miRNA and extracellular vesicles in the detection of acute and chronic SI (lines 616-619) by adding relevant references.

Reviewer 2 Report (New Reviewer)

Comment 1- In the introduction, I suggest writing a brief paragraph regarding the clinical implications of SOFA Score in diagnosis, treatment, and prognostic assessment in relation to Sepsis.

Comment 2- It would be good to briefly describe systemic inflammation, systemic inflammatory response, and chronic low‐grade inflammation.‐

Comment 3- Please include the correlation between sepsis and multiple organ dysfunction and fatality.

Comment 4- The discussion is good, but the authors should include a brief introduction of molecular mechanisms linked with systemic inflammation.

Comment 5- The significance of the present study should be included in the introduction section.

Comment 6- It is advised to include the role of inflammation in hepatic cancer.

Comment 7- The presentation of data in the form of tables is very poor. The outcomes must be presented in a better way. The ligands must be more informative and improved.

Comment 8- The conclusion should be rewritten to better present the results of the current study and future studies.

Comment 9- The graphical abstract should be provided.

Author Response

Dear Reviewer,

Thank you very much for your painstaking work, careful reading of our paper, and for your valuable comments and advices. We enclose our corrected manuscript with all highlighted changes. We enclose a response to all your comments point-by-point.

Comment 1- In the introduction, I suggest writing a brief paragraph regarding the clinical implications of SOFA Score in diagnosis, treatment, and prognostic assessment in relation to Sepsis

We have added a paragraph describing the SOFA scale in the Introduction section (lines 92-102)

Comment 2- It would be good to briefly describe systemic inflammation, systemic inflammatory response, and chronic low‐grade inflammation.

We have added a corresponding paragraph to the Introduction section (lines 43-68). However, given the large total volume of our manuscript, we were only able to focus on the most general characterization of these processes.

Comment 3- Please include the correlation between sepsis and multiple organ dysfunction and fatality.

We have added an answer to this question to the Conclusion section (lines 844-858)

Comment 4- The discussion is good, but the authors should include a brief introduction of molecular mechanisms linked with systemic inflammation.

The molecular mechanisms of proinflammatory cellular stress and different variants of inflammation are very diverse and are presented by us in two large reviews [doi: 10.3390/ijms23094596; doi: 10.2174/1381612825666190319114641]. There are References to these articles in our manuscript.  In this article, we focused only on some molecular criteria for SI, including, at the recommendation of another reviewer, we have shown the role of sTREM1, presepsin, and miRNA (lines 615-619). In addition, we have demonstrated the relationship between different variants of inflammation and cellular proinflammatory stress in the Introduction (Fig. 1) (lines 58-68). If we had to describe numerous molecular mechanisms, we would overcomplicate this already difficult to read and understand original article.

Comment 5- The significance of the present study should be included in the introduction section.

We have added a paragraph about the significance of our study in the "Introduction" section (lines 144-147)

Comment 6- It is advised to include the role of inflammation in hepatic cancer.

We have added information regarding the pathogenetic role of local and systemic proinflammatory mechanisms in the development of liver cancer (section Discussion, lines 599-601)

Comment 7- The presentation of data in the form of tables is very poor. The outcomes must be presented in a better way. The ligands must be more informative and improved.

We have edited some tables and their legends for a better understanding of the presented data (tables 1,2,4,7).

Comment 8- The conclusion should be rewritten to better present the results of the current study and future studies.

We have completely rewritten the conclusion, in which we emphasized the practical importance of this study and the prospects for further work.

Comment 9- The graphical abstract should be provided.

We agree with your opinion that the graphical representation of the results contributes to a better understanding of the work. However, the graphical abstract is not mandatory, but only a desirable section of the article, so we did not present it. It is also difficult for us to present it within the 5 days we are given to edit the paper, as this process is quite time-consuming and requires additional graphic design specialists. However, we have tried to improve our work by including an additional figure in the Introduction section. In addition, Figure 7 in the Conclusion shows the main results about the dynamics of acute SI. We thank you for your comment and will take it into account when submitting future manuscripts.

This manuscript is a resubmission of an earlier submission. The following is a list of the peer review reports and author responses from that submission.

Round 1

Reviewer 1 Report

The Authors evaluated in about 600 patients a pattern of inflammatory peptides, creating a scale of risk, to be uses primarily for scientific research. 

In the introduction section, they described the nomenclature used in the inflammation field, and I agree when they cited the error between SI and SIR. This work aims to differentiate acute and chronic SI.

Here my doubts:

Which is the target of the reader? It is very hard to read this work for a clinician.  In particular, in the section "Methodological approaches...". It seems a methodological work to highlight a process, the SI, but it is very difficult to translate these considerations in the common clinical practice. And in the reseacrh filed, which benefits a resercher could have? This method was compared with others scales?  

Are this scale influnced by chronic disease (for example chronic kidney disease, asthma, chronic heart failure)? 

 Inflammation could also be related with infection. Why did the Authors not dose for example procalcitonin. All these patients were inflammed, infected and not infected, as the authors described in the Method section. Were there differences between infected and not infected patients? How the authors exclude and comment this bias?

Are these data influenced by therapies? No mention to therapy done in these patients before the dosage on cytokines. 

Does not it represent a bias enroll 17 different types of patients and, consequently, analyse 17 different types of inflammatory pathologies? For example, how can I comment the trend of the cytokine levels in patients treated by hemodialysis vs Ankylosing spondylitis? The same doubt for  cortisol analysis.  

Reviewer 2 Report

In this paper the Authors use a classification of SI that has been proposed by the same group and has not undergone external validation / review. The objective of the study is not very clear. Also, the paper is too long and hard to understand. What is the novelty and utility of the results, if any? It is known that a higher inflammation is linked to worse outcomes. The biomarkers and cut-off chosen by the authors appear to be quite arbitrary.

Finally, the methodology appears to be flawed (i.e. it is not clear if the scales were divided using a scientific method or per opinion of the authors).